# Exploring the Therapeutic Potential of BRCA1 and BRCA2 as Targets in Canine Oncology: A Comprehensive Review of Their Role in Cancer Development and Treatment

**DOI:** 10.3390/ijms26041768

**Published:** 2025-02-19

**Authors:** Jayson Cagadas Pasaol, Agnieszka Śmieszek, Aleksandra Pawlak

**Affiliations:** Department of Pharmacology and Toxicology, Faculty of Veterinary Medicine, Wrocław University of Environmental and Life Sciences, Norwida 31, 50-375 Wrocław, Poland; jayson.pasaol@upwr.edu.pl (J.C.P.); agnieszka.smieszek@upwr.edu.pl (A.Ś.)

**Keywords:** comparative oncology, BRCA genes, canine cancer, PARP inhibitors, genomic stability

## Abstract

Tumor diseases represent a significant global health challenge, impacting both humans and companion animals, notably dogs. The parallels observed in the pathophysiology of cancer between humans and dogs underscore the importance of advancing comparative oncology and translational research methodologies. Furthermore, dogs serve as valuable models for human cancer research due to shared environments, genetics, and treatment responses. In particular, breast cancer gene 1 (BRCA1) and breast cancer gene 2 (BRCA2), which are critical in human cancer, also influence the development and progression of canine tumors. The role of BRCA1 and BRCA2 in canine cancers remains underexplored, but its potential significance as therapeutic targets is strongly considered. This systematic review aims to broaden the discussion of BRCA1 and BRCA2 beyond mammary tumors, exploring their implications in various canine cancers. By emphasizing the shared genetic underpinnings between species and advocating for a comparative approach, the review indicates the potential of BRCA genes as targets for innovative cancer therapies in dogs, contributing to advances in human and veterinary oncology.

## 1. Introduction

Cancer, characterized by uncontrolled growth and proliferation, significantly affects health systems worldwide [1]. Despite the transient change in mortality patterns due to COVID-19 in 2020, cancer consistently ranks second as the leading cause of death, second only to cardiovascular diseases in the United States and Europe [2,3]. This global health problem extends beyond humans [4], as cancer is also recognized as the leading cause of disease-related mortality in dogs in developed countries [5,6,7,8,9,10,11]. As of 2023, the global dog population is estimated at 900 million [12], with approximately 470 million living as pets [13]. Retrospective studies indicate that cancer accounts for about 30% of all canine deaths [10,14], underscoring its prevalence and impact on the canine population.

The shared environments and lifestyles between dogs and their human companions have facilitated the emergence of dogs as valuable models for understanding human cancers. The spontaneous development of dog cancers, similar to those in humans, along with similar responses to treatment, positions canine cancer research as a crucial parallel to human oncological studies [15]. In comparative oncology, canine tumor models have unique advantages over other animal models, particularly rodents. Dogs naturally develop spontaneous tumors that closely mimic human cancers in their histopathology, genetic mutations, tumor microenvironment, and treatment response. This spontaneous tumorigenesis mirrors the progression of human cancer more accurately than the induced models in rodents [16]. Furthermore, dogs share common environmental exposures and lifestyle factors with humans, increasing their relevance in translational research. The shorter lifespan of dogs also allows for the rapid assessment of long-term therapeutic outcomes and side effects, facilitating faster clinical translation [17].

Hence, the field of comparative oncology has recognized that naturally occurring cancers in dogs offer invaluable information on human oncology, especially given the common pathophysiology of these diseases and their similarity in risk factors, tumor biology, and response to treatment. Recent genomic studies have highlighted significant similarities between canine and human cancers, revealing common genetic underpinnings and providing information on tumor development between species. In particular, cancer types such as osteosarcoma, melanoma, non-Hodgkin lymphoma, and bladder cancer share histological and molecular characteristics between dogs and humans, suggesting a shared oncogenic pathway and potential for comparative oncology research [18,19,20].

Among the various cancers that affect dogs in Europe, canine mammary tumors (CMTs) are particularly prevalent, especially in uncastrated female dogs [21,22,23]. Research has identified candidate genes associated with CMT development, including breast cancer gene 1 (BRCA1) and breast cancer gene 2 (BRCA2) [24,25]. These genes, known for their role in human breast and ovarian cancers, have also been implicated in CMTs, highlighting the genetic similarities between human and canine cancers [24,26,27]. However, its role in canine cancer remains underexplored despite the established importance of BRCA1 and BRCA2 as biomarkers and therapeutic targets in human oncology.

This review aims to expand the discussion of BRCA1 and BRCA2 beyond their association with CMTs, exploring their broader implications in the oncogenesis of various dog cancers. By highlighting the frequency and impact of cancer in dogs, drawing parallels with human oncology, and delving into the genetic underpinnings shared between species, this review emphasizes the potential of BRCA genes as targets for innovative cancer therapies. Furthermore, it underscores the importance of a comparative approach, recognizing the value of canine models in elucidating the molecular mechanisms of cancer and in the development of targeted treatments.

## 2. Methods

A systematic literature search from 1990 to 2023 was performed in PubMed and the Cochrane Library to ensure a comprehensive and current overview. Search terms included “canine mammary tumors”, “BRCA1 and BRCA2”, “PARPi”, and “DNA repair mechanisms”. This methodological approach facilitated the inclusion of relevant clinical trials, reviews, meta-analyses, and other pivotal studies that supported the review investigation into the therapeutic importance of BRCA1 and BRCA2 as potential targets in canine cancer. The search followed the PRISMA (Preferred Reporting Items for Systematic Reviews and Meta-Analyses) guidelines to ensure transparency and reproducibility. A total of 250 studies were identified, of which 120 were excluded according to relevance, leaving 130 studies for detailed analysis. However, only 95 studies were directly cited in this review, as they provided the most pertinent data to support the investigation.

## 3. BRCA1 and BRCA2 in Canine Cancer: Mechanisms and Therapeutic Implications

### 3.1. BRCA1 and BRCA2: Structural, Biological, and Molecular Functions

#### The Role of BRCA1 and BRCA2

BRCA1 (Figure 1A) and BRCA2 (Figure 1B) are pivotal tumor suppressor genes, each playing a significant role in crucial cell processes for maintaining genomic stability. BRCA1 encodes proteins involved in various cellular processes, including DNA repair mechanisms, particularly through homologous recombination (HR), essential to ensure genomic stability [28]. Beyond its role in DNA repair, BRCA1 also regulates cell cycle progression and transcriptional control, influencing cell growth and division, thus preventing abnormal cell proliferation [29]. Similarly, BRCA2 is also involved in DNA repair pathways. Its protein product primarily facilitates HR by interacting with other proteins to repair DNA double-strand breaks (DSB), a critical function for maintaining genomic integrity [30]. Together, BRCA1 and BRCA2 cooperate in a concerted effort to repair damaged DNA, regulate cell growth, and prevent cancer development by ensuring that the cell’s genetic material remains stable and intact.

### 3.2. Molecular and Biological Functions of BRCA1 and BRCA2 in Cancer: A Comparative View Between Humans and Canines

In humans, mutations in the BRCA1 and BRCA2 genes are strongly associated with an increased risk of breast, ovarian, prostate, pancreatic, melanoma, and peritoneal cancers, highlighting their essential role in DNA repair mechanisms, cell cycle control, and tumor suppression [31,32,33,34,35]. Although they are located on different chromosomes, both genes share similar functions in DNA damage repair, transcriptional regulation, and genomic integrity [34,35], thus preventing the onset of cancer.

In canines, the BRCA1 and BRCA2 genes share structural and functional homology with their human counterparts (Figure 2), indicating their potential importance in comparative oncology. Recent studies have begun to uncover the significance of these genes in dogs, particularly their association with mammary tumors [23,24]. Germline mutations in canine BRCA1 and BRCA2 have been associated with a higher risk of CMTs [36], analogously to their role in human breast and ovarian cancers. However, research on these mutations and their functional implications in dogs is still evolving, with findings that signify similarities and unique aspects of canine cancer pathogenesis. Expression analyses of BRCA1 and BRCA2 in CMTs reveal complex interactions with DNA repair proteins such as RAD51 recombinase (RAD51) [36,37], reflecting their human functions and indicating species-specific regulatory mechanisms. For example, overexpression of RAD51 in canine tumors suggests a distinct regulatory loop that may differ from human cancer pathways.

Unlike the human context, where BRCA1 expression is a strong indicator of malignancy, canine oncology presents a more complex landscape. Recent advances have shown a distinctive expression pattern involving the genes Bone Morphogenetic Protein 2 (BMP2), Latent Transforming Growth Factor Beta Binding Protein 4 (LTBP4), and Derlin 1 (DERL1) in CMTs. This intricate signature of mRNA expression offers a promising malignancy pattern in dogs, highlighting the diversity of cancer genetics between species [38,39]. Dogs exhibit remarkable genetic diversity between breeds, significantly influencing cancer biology, mutation prevalence, and treatment responses. Due to inherited genetic mutations, certain breeds, such as Boxers, Golden Retrievers, and Cocker Spaniels, exhibit a higher predisposition to specific cancer types, including mammary tumors and lymphomas [24]. This genetic variability affects the function and frequency of the BRCA1 and BRCA2 genes, which could alter the sensitivity of different breeds to poly [ADP-ribose] polymerase inhibitors (PARPi) and other targeted therapies. For example, breed-specific variations in DNA repair pathways may affect the efficacy and toxicity profiles of BRCA-targeted treatments. Therefore, understanding these breed-related genetic differences is crucial to optimize therapeutic outcomes in canine oncology. Future research should prioritize comprehensive breed-specific genetic studies to tailor personalized treatment approaches for dogs with BRCA-associated cancers [27,40].

The characterization of BRCA1 and BRCA2 in canine cancers remains inconsistent between existing studies. This inconsistency arises from varying methodologies, different sample sizes, and the lack of standardized reporting protocols. As a result, it is challenging to draw definitive conclusions about the prevalence, functional significance, and therapeutic implications of BRCA mutations in canine cancer. Although some studies have identified BRCA mutations associated with mammary tumors and other cancers in dogs, variations in sequencing technologies and criteria for variant classification contribute to conflicting findings [24,27]. To address these discrepancies, there is an urgent need for standardized genetic testing protocols, unified reporting frameworks, and multi-institutional collaborations that can harmonize research efforts. Comparative genomic analyses, which evaluate BRCA variants in larger canine populations and multiple types of cancer, could improve the reliability and reproducibility of the findings. Establishing consistent methodologies will be crucial to advancing precision medicine in veterinary oncology and ensuring that BRCA-targeted therapies are accurately tailored to canine patients.

Such insights are crucial because they expand our understanding of tumor biology in dogs, potentially guiding the application of targeted therapies such as PARPi. Detailing the comparative analysis of the BRCA1 and BRCA2 genes in humans and dogs indicates these genes’ critical roles in maintaining genomic integrity and preventing cancer in species. Table 1 illustrates not only the genetic locations of these genes, BRCA1 on human chromosome 17q21 and canine chromosome 9, and BRCA2 on human chromosome 13q13.2 and canine chromosome 25, but also their substantial genetic lengths, which in humans translates to a protein comprising 1863 amino acids for BRCA1 and 3418 for BRCA2.

The table further reveals these genes’ high degree of evolutionary conservation, highlighting their fundamental role in cellular defense mechanisms. For example, BRCA1 is highly conserved between humans and dogs, suggesting a shared mechanism of tumor suppression and DNA repair, as evidenced by the crucial RING and BRCT domains [34,41,42,43,44]. Similarly, conservation is reflected in its BRC repeats and helicase domains, indicating a conserved function in DNA repair [27,45,46,47,48,49,50].

Evaluating the differences in gene expression in various tumor types is critical to elucidate further the functional implications of BRCA1 and BRCA2 in canine tumors. These analyses improve our understanding of their roles in tumorigenesis and underscore their potential as therapeutic targets, particularly in targeted treatments such as PARPi. Techniques such as quantitative real-time PCR (qPCR) and Western blot enable the precise quantification of BRCA1 and BRCA2 at both the mRNA and protein levels. At the same time, immunohistochemistry (IHC) offers valuable insight into its localization and differential expression within tumor tissues. Furthermore, advanced methods such as RNA sequencing (RNA-seq) and next-generation sequencing (NGS) provide comprehensive genetic profiles, identifying mutations and expression patterns that can mirror or diverge from those observed in human cancers [27,40,51].

Comparative genomic hybridization (CGH) and functional assays, such as RAD51 foci formation tests, further elucidate DNA repair deficiencies that contribute to the development of canine tumors [52]. These approaches are essential to bridge the knowledge gap between human and canine oncology, encourage the development of comparative therapeutic strategies, and refine breed-specific treatment protocols to improve clinical outcomes in veterinary medicine.
ijms-26-01768-t001_Table 1Table 1Comparative analysis of the BRCA1 and BRCA2 genes in humans and dogs.
BRCA1 (Human)BRCA1 (Canine)BRCA2 (Human)BRCA2 (Canine)Chromosome Location17q21Chr.913q12.3Chr.25Gene Length (base pairs)~100,000~85,000~84,000~81,000Protein Length (amino acids)1863183234183414Conservation across speciesHighly conservedConservedHighly conservedConservedBiological FunctionTumor suppressorTumor suppressorDNA repairDNA repairProtein DomainsRING, BRCT, SQ/TQ clusterRING, BRCT, SQ/TQ clusterBRC repeats, DNA-binding, helicaseBRC repeats, DNA-binding, helicaseAssociation with CancerBreast, ovarian, other cancersMammary gland,  ovarian, other cancersBreast, ovarian, other cancersMammary gland, ovarian, other cancersGenetic VariantsNumerous pathogenic variants identifiedLimited studies on  genetic variantsNumerous pathogenic variants identifiedLimited studies on genetic variantsMutational SpectrumPoint mutations,  insertions, deletionsPoint mutations,  insertions, deletionsPoint mutations,  insertions, deletionsPoint mutations,  insertions, deletionsDisease RiskIncreased risk of breast, ovarian, and other breast cancer  cancersIncreased risk of infection  mammary gland,  ovarian, and other  cancersIncreased risk of breast, ovarian, and other breast cancer  cancersIncreased risk of mammary gland, ovarian, and other cancersReferences[34,41,42][43,44,51][45,46,47,48][27,49,50]


## 4. Implications of BRCA1 and BRCA2 in Cancer Therapy

### 4.1. Targeting BRCA1 and BRCA2 Using PARP Inhibitors (PARPi) for Personalized Therapies

The therapeutic landscape of cancer treatment has been profoundly influenced by targeting BRCA1 and BRCA2 mutations, leading to innovative approaches and advancements in cancer therapy. These genetic markers have paved the way for personalized medicine in oncology, primarily through the application of PARPi. Specifically, the mechanism of action of PARPi targets the enzyme PARP (Poly [ADP-ribose] polymerase), which is crucial for repairing single-strand DNA breaks. By inhibiting PARP, these drugs prevent DNA damage repair, leading to the accumulation of DNA breaks. In cells with BRCA1 or BRCA2 mutations, which cannot repair double-strand breaks already by homologous recombination, this accumulation of DNA damage becomes lethal, causing cancer cell death [53,54,55,56,57,58,59,60,61]. The groundbreaking aspect of using olaparib in human medicine for treating cancers with BRCA1 and BRCA2 mutations was demonstrated in patients with metastatic breast tumors that harbor BRCA1/2 mutations, achieving an overall response rate (OR) of 50%. This approach exploited the absence of homologous DNA repair in cancer cells [62].

In the veterinary field, companies like Vidium Animal Health, a subsidiary of the Translational Genomics Research Institute, have used comprehensive NGS panels to identify genomic mutations in dogs diagnosed with various cancers. This panel facilitates the identification of genetic aberrations such as single nucleotide variants, insertions or deletions, copy number variants, and internal tandem duplications across a broad spectrum of cancer-related genes. For example, it can detect BRCA1/BRCA2 mutations associated with breast and ovarian cancer risk, followed by mutations in genes such as TP53, known for its role in cell cycle regulation and apoptosis, and EGFR, which is relevant in developing non-small cell lung cancer. This comprehensive strategy shows its ability to cover various genetic variations involved in oncogenesis [63]. A recent study collected clinical data and analyzed 336 unique mutations in 89 genes in 26 types of cancer. The study found that mutations in six specific genes such as Cyclin D1 [CCND1], Cyclin D3 [CCND3], SWI/SNF Related, Matrix Associated, Actin Dependent Regulator of Chromatin Subfamily B Member 1 [SMARCB1], Fanconi Anemia Complementation Group G [FANCG], Cyclin-Dependent Kinase Inhibitor 2A/2B [CDKN2A/B], and MutS Homolog 6 [MSH6] were significantly associated with shorter progression-free survival (PFS) in canine patients [64].

Furthermore, the research highlighted the potential benefits of using olaparib as a targeted treatment. Dogs that received targeted therapy before their first cancer progression (*n* = 45) experienced significantly longer PFS than those who did not receive such treatments (*n* = 82, *p* = 0.01). This benefit was also observed in a subset of 29 dogs that received genomically informed targeted treatment, showing a statistically significant improvement in PFS compared to those who did not receive such personalized treatments (*p* = 0.05) [65]. These findings emphasize the importance of genomic analysis and precision medicine in veterinary oncology, demonstrating that dogs with cancer can benefit from targeted therapies based on specific genomic mutations.

The reported clinical trials with PARPi and their observed efficacy in various types of cancer in humans (Table 2) provide a reference for the possible translational impact these findings may have on the treatment of canine cancers. Experience in human oncology, including developing resistance to PARPi and strategies to overcome it [66,67,68], shows that these drugs in humans and dogs also improve our understanding of the molecular underpinnings of different types of cancer. They may also reveal novel therapeutic targets and lead to the development of new treatment regimens, which could include combination therapies involving PARPi [69] and immune checkpoint inhibitors or other agents that target DNA damage response [70,71].

Interestingly, BRCA mutations are not the only indication for the use of PARPi. The possibility of effective use of PARPi in treating cancer without BRCA mutations is related to the appearance of a phenomenon known as ‘BRCAness’. The concept refers to the phenotype observed in certain tumors. Despite the lack of mutations in the BRCA1 or BRCA2 genes, these tumors exhibit similar functional deficiencies in DNA repair mechanisms, particularly in the HR pathway [55]. This phenomenon is crucial in humans and dogs, as it influences the response to specific therapies, especially PARPi. Deficits typically involve genes other than BRCA1/2 but are essential for effective HR, such as ATM, ATR, PALB2, and RAD51 [80,81,82]. These genes are crucial in detecting, signaling, and repairing DNA double-strand breaks. Their dysfunction leads to a characteristic inability to repair DNA properly, similar to that seen with BRCA mutations.

A multi-faceted approach is often used to detect ‘BRCAness’ in canine cancers, combining genetic, genomic, and functional assays. Genetic screening can be performed through NGS to identify gene mutations associated with HR beyond BRCA1/2 [83]. Genomic approaches, such as CGH, can detect genomic scars, deletion patterns, duplications, and rearrangements indicative of a history of defective DNA repair [84]. Furthermore, functional assays, such as the RAD51/ℽH2Ax foci formation assay, directly assess the ability of tumor cells to mount an effective DNA repair response after induced DNA damage [85]. Through these methods, veterinarians can pinpoint canine tumors that, although negative for the BRCA mutation, exhibit a profile of ‘BRCAness’, thus extending the potential application of PARPi therapy.

Beyond BRCA1 and BRCA2, several other genes involved in DNA repair and cell cycle regulation merit investigation in canine oncology. Tumor suppressor genes, such as tumor protein 53 (TP53), frequently mutated in various cancers, play a crucial role in maintaining genomic stability. Mutations in the phosphatase and tensin homolog (PTEN), a key regulator of cell proliferation and apoptosis, are also of significant interest because of their role in tumorigenesis. Furthermore, genes associated with homologous recombination repair, such as mutant ataxia Telangiectasia (ATM), Ataxia Telangiectasia and Rad3 related (ATR), Partner and Localizer of BRCA2 (PALB2) and RAD51, contribute to critical DNA repair processes. These genes may exhibit ‘BRCAness’ phenotypes characterized by deficiencies in homologous recombination repair, even in the absence of BRCA mutations. Collectively, these genes represent potential targets for PARPi and other therapeutic strategies, thus expanding the scope of precision medicine in veterinary oncology [64].

The cross-application of therapeutic strategies between human and veterinary oncology, particularly for dogs with cancer associated with BRCA mutations and other types of cancers, offers a promising path to improve treatment outcomes. Establishing reliable diagnostics and expanding the genetic focus beyond BRCA mutations will be critical for optimizing the use of PARPi and advancing precision medicine in canine oncology.

Although PARPi has shown significant efficacy as monotherapy in both human and canine oncology, emerging research highlights the potential of combination therapies to improve treatment outcomes and overcome resistance mechanisms. In human oncology, combining PARPi with agents that induce DNA damage, such as platinum-based chemotherapies, has synergistic effects, leveraging complementary pathways to induce cancer cell death [86]. Similarly, immune checkpoint inhibitors promote antitumor immune responses and have shown increased efficacy when combined with PARPi due to up-regulation of PD-L1 expression and increased immune activation [87]. Furthermore, hypomethylating agents, which modulate gene expression, sensitize tumors to PARPi by affecting DNA repair pathways, offering another promising combination strategy [69].

These combination strategies are highly relevant for canine cancer, where the biological behavior of cancer often mirrors that of human cancers. Given the shared environmental factors and genetic mutations, canine models provide an excellent platform to assess the efficacy of these combination therapies. Future studies should focus on identifying optimal dosing regimens, possible side effects, and biomarkers for predicting responses to combination treatments in dogs. Table 3 summarizes the key combination strategies involving PARPi and potential translational applications in veterinary oncology.

### 4.2. Advancing BRCA-Targeted Therapies in Veterinary Oncology

Little information is available in the literature on the mechanism of action and the possible use of PARPi in dogs. Existing in vitro experiments have laid the groundwork for demonstrating the susceptibility of canine cancer cells with deficient DNA repair mechanisms to PARPi, mirroring the success observed in human cancer cells. For example, our recent study explored the effects of olaparib on specific canine lymphoma and leukemia cell lines, namely CLBL-1 and GL-1. This study aimed to evaluate whether olaparib could exploit vulnerabilities within the DNA repair mechanisms of these cancer cells, similar to its proven success in human oncology. In our study, we used a series of in vitro assays to assess the impact of olaparib on canine lymphoma and leukemia cell lines using the MTT assay for cell proliferation, flow cytometry to analyze cell cycle progression and apoptosis, and detection of histone-phosphorylated H2AX to indicate DNA damage. Significant results showed that olaparib markedly inhibited the proliferation of tested canine cancer cell lines in both concentration and time-dependent manners. This inhibition was characterized by a substantial arrest of cells in the G2/M phase of the cell cycle, together with an increase in apoptosis. Such results suggest that the efficacy of olaparib in inducing synthetic lethality likely arises from its interference with DNA repair pathways, exploiting intrinsic deficiencies within cell repair mechanisms [90].

Although studies on the use of PARPi in dogs are limited, the application of olaparib in veterinary oncology is increasing, supported by genetic analysis results. For example, a case study highlighted the effectiveness of olaparib, administered at a dose of 3 milligrams per kilogram of dog body weight daily, in treating a spayed female pug diagnosed with malignant oral melanoma [91]. Another method was implemented, such as FidoCure DNA sequencing, which identified ATM and BRCA1 mutations, leading to using olaparib as a potential treatment. This treatment resulted in significant positive changes, including a reduction in the size of the affected lymph nodes and a noticeable shrinkage of the tumor mass [91]. Similarly, another clinical study, facilitated by FidoCure^®^, involved a 13-year-old Shih Tzu diagnosed with adrenocortical carcinoma. The patient’s medical journey began in June 2020 with symptoms that led to the discovery of an adrenal mass, which was subsequently diagnosed after surgical removal and chemotherapy treatments. Despite initial efforts, the patient experienced local recurrence and metastasis, including a significant liver mass identified as metastatic adrenocortical carcinoma. In a shift towards precision medicine, FidoCure^®^ DNA sequencing revealed mutations in BRCA1 and PTEN within the tumor, indicating that targeted therapies with PARPi olaparib and mTOR inhibitor rapamycin could be beneficial and effective [92].

From May 2023, olaparib was administered, showing no adverse effects and significant clinical improvement, including reduced liver metastasis and stabilization of the adrenal gland mass [92]. These results emphasize the potential of precision medicine in veterinary oncology, particularly in managing complex and advanced-stage cancers. These studies underscore the importance of genetic testing in identifying targeted therapies that can lead to better outcomes for pets with cancer.

Despite the promise of BRCA-targeted therapies in canine cancer, several challenges must be addressed. One of the main obstacles is the limited understanding of the prevalence and functional implications of BRCA mutations in different canine cancers. Although BRCA1 and BRCA2 have been associated with CMTs, their roles in other types remain underexplored [27]. Furthermore, the absence of standardized diagnostic tools to detect BRCA mutations and ‘BRCAness’ phenotypes in dogs complicates the identification of suitable candidates for targeted therapies. Influenced by genetic diversity, the pharmacokinetic and pharmacodynamic differences between dog breeds further challenge the optimization of drug dosages and treatment regimens [93]. Furthermore, the risk of resistance to PARPi, a common challenge observed in human oncology, can pose a hurdle in veterinary applications and warrants further investigation [94].

However, determining the optimal dose of PARPi [95] is crucial, particularly in dogs with significant genetic diversity between breeds, to maximize therapeutic efficacy while minimizing potential side effects. However, long-term studies are also essential to evaluate the safety and overall benefits in the veterinary setting.

The potential of PARPi and combination therapies [88], specifically in canine cancer treatment, highlights a significant advance in veterinary oncology, drawing parallels to their successful application in human cancer treatment. The move toward more personalized medicine in dogs is expected to improve treatment outcomes and provide a deeper understanding of cancer biology that transcends species boundaries. Establishing reliable diagnostics, understanding breed-specific genetic predispositions, and fostering interdisciplinary collaborations are essential in integrating these advanced therapies into daily veterinary care. Ultimately, these efforts aim to improve the quality of life of dogs diagnosed with cancer and contribute to the larger fight against this complex disease.

## 5. Conclusions and Future Directions

Evidence suggests that BRCA1 and BRCA2 are crucial targets for innovative canine cancer therapies, offering promising parallels to human medicine and underlining the importance of a comparative approach. Despite findings on evolutionary conservation and functional similarities between species, the different aspects of canine cancer biology and genetics need further investigation. Although preclinical studies and case reports provide foundational information on the therapeutic potential of BRCA1 and BRCA2 in canine oncology, the lack of large-scale clinical trials remains a significant limitation. The current body of evidence is based primarily on in vitro studies and isolated clinical cases, which may not fully capture the diversity of canine cancers or accurately predict PARPi treatment responses between different breeds and tumor types. Furthermore, long-term follow-up data are scarce, limiting our understanding of the durability of treatment responses, the emergence of resistance mechanisms, and potential chronic side effects. Given the genetic diversity among dog breeds, which can influence BRCA1/2 function and mutation prevalence, breed-specific studies are essential to tailor therapeutic approaches effectively [24,27].

Future research should expand our understanding of BRCA mutations in various canine cancers, improve diagnostic techniques, investigate ‘BRCAness’ in canine cancers, and develop targeted therapies, including PARPi. Furthermore, while PARPi monotherapy has shown promise, emerging evidence from human cancer suggests that combination therapies—such as PARPi with platinum-based agents, immune checkpoint inhibitors, or hypomethylating agents—may enhance efficacy and delay resistance [69,88]. Future research should focus on standardized genetic characterization in canine studies and explore these combination strategies in veterinary clinical trials to optimize treatment outcomes and advance precision medicine in canine oncology.

This approach may improve the better diagnosis, treatment outcomes, and prognosis of cancer. The significance of this research extends beyond canine oncology, enriching the understanding of comparative oncology, offering novel insights into cancer biology, and fostering advances in both veterinary and human medicine.

## Figures and Tables

**Figure 1 ijms-26-01768-f001:**
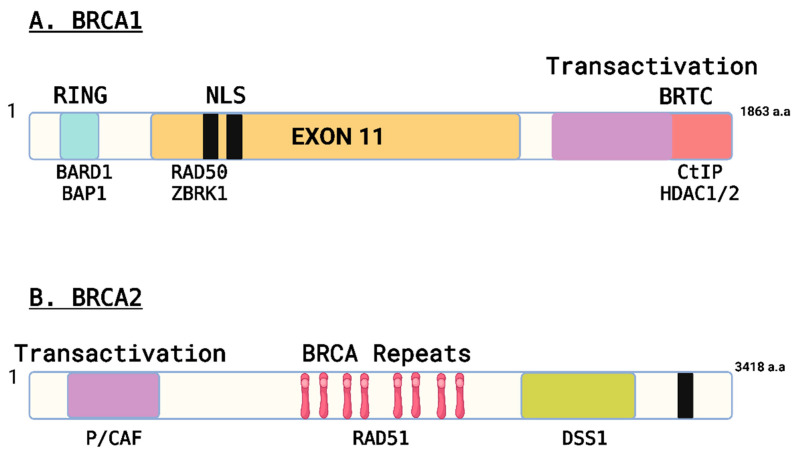
Structural and functional motifs recognized in BRCA1 and BRCA2. (**A**) Schematic structure of BRCA1. The RING domain, the transactivation domain, the BRCT domain, nuclear localization signals (NLS), and exon 11 coding regions are indicated. Representative proteins that interact with BRCA1 in three different regions are shown. (**B**) Schematic structure of BRCA2. The transactivation domain, the BRC repeats, and the NLS are indicated. Representative BRCA2-interacting proteins are indicated.

**Figure 2 ijms-26-01768-f002:**
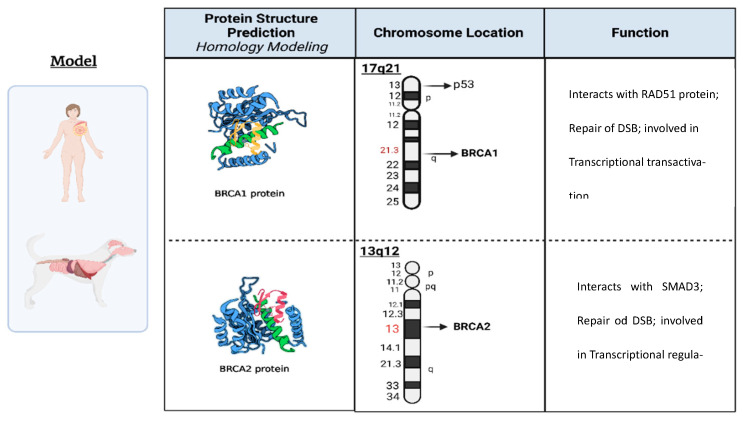
Summary of essential information for BRCA1 and BRCA2: 3D protein structure, chromosomal location, and primary function.

**Table 2 ijms-26-01768-t002:** Synopsis of reported clinical trials involving PARPi.

PARP Inhibitor Tested	Cancer Type	Efficacy	Refs
Olaparib	Solid tumors (ovarian: 35%)	Clinical benefit for 63% (in patients with carriers of carriers of patients with BRCA mutations carriers’ patients)	[72]
	Breast	ORR*: 41%	[73]
	Ovarian	ORR*: 33%	[74]
	Ovarian, breast, pancreatic and prostate	Tumor response rate*: 26.2%	[75]
Rucaparib (Temozolomide)	Metastatic melanoma	Clinical benefit for 34.8% of the patients	[76]
	Advanced solid malignancies	CR: 1/32 (melanoma) PR: 2/32 (melanoma, desmoid tumor)	[77]
Veliparib	Metastatic breast cancer	ORR (CR+PR) 12.5% (3/24)	[78]
Iniparib	Metastatic TNBC	ORR (CR+PR) 32%	[79]

ORR* (Objective Response Rate): according to RECIST, with confirmation of response at least 28 days apart by CT and RECIST. Tumor response rate*: according to RECIST, with confirmation of the response at least 28 days apart; CR—complete response; PR—partial response.

**Table 3 ijms-26-01768-t003:** Summary of PARP inhibitor combination therapies in human and canine oncology.

Combination Therapy	Mechanism of Action	Evidence in Human Oncology	Potential Application in Canine Model
PARPi + Platinum-Based Chemotherapy	Both agents exploit DNA repair deficiencies, leading to enhanced DNA damage and apoptosis.	Synergistic effects on ovarian and breast cancers [88].	Promising for CMTs and lymphomas.
PARPi + Immune Checkpoint Inhibitors	Enhance antitumor immunity by upregulating PD-L1 and promoting the immune response.	Improved response rates in various cancers [87].	Potential for canine melanomas and lymphomas.
PARPi + Hypo- methylating Agents	Epigenetic modulation sensitizes tumors to PARPi by affecting DNA repair pathways.	Effective in breast and ovarian cancers [69].	Investigated for canine mammary and hematopoietic cancers.
PARPi + Radiation Therapy	Radiation induces DNA double-strand breaks, complemented by PARPi-induced inhibition of DNA repair mechanisms.	Enhanced tumor control in glioblastoma and prostate cancers [89].	Could improve outcomes in canine osteosarcoma and mast cell tumors.

## Data Availability

According to the review nature of the manuscript, all sources are open and available in the references.

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
