# Peer review of "Exploring the Therapeutic Potential of BRCA1 and BRCA2 as Targets in Canine Oncology: A Comprehensive Review of Their Role in Cancer Development and Treatment"

_ijms, 2025, doi:10.3390/ijms26041768_

Round 1
Reviewer 1 Report
Comments and Suggestions for Authors
Jayson. et al. Oncologic disease is a common health challenge for humans and dogs, with BRCA1 and BRCA2 genes playing a key role in tumor development in both species. By comparing the common genetic basis of canine and human tumors, especially the role of BRCA gene, it can provide new ideas and opportunities for cancer therapy research and development in the two species. Moreover, the result is technically sounded and worthy to be published in Int. J. Mol. Sci.
The following are some comments and suggestions that are given to improve the manuscript:
Comment 1: How to evaluate the expression difference of BRCA gene in different canine tumors?
Comment 2: What are the unique advantages of canine tumor models over other animal models?
Comment 3: What are the potential challenges of BRCA gene-targeted therapy in the treatment of canine tumors?
Comment 4: In addition to the BRCA gene, which common genes are worth studying?

Reviewer 2 Report
Comments and Suggestions for Authors
While this review article provides a comprehensive overview of the therapeutic potential of BRCA1 and BRCA2 in canine oncology, there are several limitations that should be noted:
1. The review relies heavily on preclinical studies and a small number of clinical cases, which may not fully represent the diverse range of canine cancers and their responses to PARP inhibitors (PARPi). The lack of large-scale clinical trials in dogs limits the robustness of the conclusions drawn regarding the efficacy and safety of BRCA-targeted therapies. In addition, the review highlights the potential benefits of PARPi in treating canine cancers but lacks long-term follow-up data. Long-term studies are crucial to assess the durability of treatment responses, the emergence of resistance mechanisms, and the potential for chronic side effects.
2. Dogs exhibit significant genetic diversity across breeds, which can influence cancer biology and treatment response. The review does not adequately address how breed-specific genetic variations might affect BRCA1/2 function, mutation prevalence, or therapeutic outcomes.
3. The review focuses primarily on PARPi as monotherapy, with limited discussion of combination strategies. A more in-depth exploration of these combinations in canine models could provide additional insights and improve treatment outcomes.
4. The review notes that studies on BRCA1/2 genetic variants in dogs are limited, and the reported variants are not consistently characterized across studies. This inconsistency makes it difficult to draw definitive conclusions.
Comments on the Quality of English LanguageOverall, the article appears to be well-written and professionally structured, with clear language and appropriate scientific terminology. However, there are a few minor language issues that could be addressed to improve clarity and readability.
Example: In Section 3.3, the phrase "The therapeutic landscape of cancer treatment has been profoundly influenced by recognizing the BRCA1 and BRCA2 mutations and targeting them" is somewhat redundant. The idea of targeting BRCA mutations is mentioned multiple times in close proximity.
Example: The term "canine mammary tumors" (CMT) is used interchangeably with "mammary tumors in dogs." While both are correct, consistent terminology would improve clarity.
Suggestion: Choose one term (e.g., CMT) and use it consistently throughout the article.
In Section 3.4, the sentence "Despite the small number of studies on the possibility of using PARPi in dogs, olaprib is increasingly used in veterinary oncology, based on the results of genetic analyses" is somewhat convoluted.
In Section 3.4, the sentence "Our team employed a series of in vitro assays to assess the impact of olaparib" does not clearly specify which team or study is being referred to.
Reviewer 3 Report
Comments and Suggestions for Authors
Dear Authors,
I appreciate the opportunity to review an interesting article covering therapeutic strategies between human and veterinary oncology. It is a reasonable source of knowledge based on up-to-date scientific reports, contributes to a better understanding of cancer biology (even in different species) and it may also inspire researchers for developing novel therapeutic approaches for the treatment of cancer both in human and veterinary care.
Some minor comments regarding your manuscript are listed below:
1. I suggest changing the reference style. In the present shape it is very difficult to find the cited article, especially in a multitude of cited papers in the review article. Especially when the reference list does not correspond with the inserted citations;
2. Some typos should be corrected throughout the manuscript (e.g.line 38, line 236) ;
3. Line 65. Please explain the abbreviation when used for the first time;
4. I know that is obvious, but please elaborate on the acronym PARP used in your manuscript;
5. In the introduction you mention a systematic review as the type of your article. However, the methodology is insufficiently described. Is your literature review founded on PRISMA guidelines? What are the exclusion criteria? How many papers were found and how many were excluded?
Best regards,
The reviewer.
Round 2
Reviewer 1 Report
Comments and Suggestions for Authors
The authors have addressed all the questions
Reviewer 2 Report
Comments and Suggestions for Authors
The manuscript was considerably improved.